# Angular and Spectral Bandwidth Considerations in BRDF Measurements of Interference- and Diffraction-Based Coatings

**Alejandro Ferrero ***[ID]  **and Joaquín Campos**[ID]

Instituto de Óptica "Daza de Valdés", Consejo Superior de Investigaciones Científicas (CSIC), C/ Serrano 144, 28006 Madrid, Spain; joaquin.campos@csic.es

*   Correspondence: alejandro.ferrero@csic.es

**Abstract:** The Bidirectional Reflectance Distribution Function (BRDF) of iridescent (or goniochromatic) surfaces may vary notably with both spectral and angular variables, and, therefore, finite spectral bandwidth and collection solid angles inherent to any measuring instrument introduce a deviation from the correct value. Experimental data of highly goniochromatic samples are used to analyse their impact on measurement uncertainty. The results indicate that it is advisable to standardize spectral and angular bandwidths because the systematic error is not negligible for typical measuring systems. The 95th percentile of the error distribution of the measurement of the BRDF due to these finite bandwidths, and also the 95th percentile of the calculated resulting color differences, are used as criteria to establish recommended values of spectral and angular bandwidths. The impact of the bandwidth is more critical in the measurement of the BRDF of diffraction-based than of interference-based coatings.

**Keywords:** BRDF measurements; goniochromatism; special effect coatings; colour

## 1. Introduction

The Bidirectional Reflectance Distribution Function (BRDF) is a quantity describing the variation of the reflectance for any combination of irradiation and reflection directions [1], and it allows the reflectance to be calculated for any irradiation and collection solid angle. The spectral BRDF, which includes in addition spectral variation, is a key quantity to describe the goniochromatism of iridescent surfaces, as those with special effect coatings [2–4] which are widely used in automotive [5], cosmetic or packaging sectors, among others. The spectral BRDF of goniochromatic surfaces changes in great extent for different combinations of illumination and viewing directions (geometries), and it makes their colorimetric description complex. Most of the research work on the appearance of goniochromatic surfaces has dealt with the proper methodology to describe their color by selecting adequate geometries, and to use measurements for representing such surfaces [6–17]. However , the metrological aspect of measuring the spectral BRDF and the color with real instruments, which have finite-size apertures, has not been systematically studied yet as in this work. Since the appearance of goniochromatic surfaces notably depends on both spectral and angular variables, the non-infinitesimal measurement spectral bandwidth and the non-infinitesimal size of the measurement collection solid angles introduce a deviation from the correct value, and it might be non-negligible in some measuring systems. Moreover, there is an interrelation between these non-finite-size effects. The non-infinitesimal solid angles impact not only on the measured angular distribution but affect also the measured spectral distribution. And vice-versa, the non-infinitesimal spectral bandwidth affects not only on the measured spectral distribution but also on the measured angular distribution. The proper understanding of the

uncertainty due to this cause is not only important for a good measurement, but also to improve the design of multiangle- [18,19] or gonio-spectrophotometers [20–29] devoted to control the appearance of iridescent surfaces. The objective of this work is to develop a methodology relating the spectral bandwidth and the collection solid angles with the resulting measurement uncertainty, and to provide guidance on the suitable or upper limit values of these variables.

The goniochromatism presented in special effect coatings can be produced by interference or diffraction pigments embedded in a binder, and its description is different for each case.

Interference pigments consist of two or more layers with a high index of refraction difference. Multiple reflections at the layer boundaries are followed by interference of the light waves. This interference, giving rise to additive color mixing, is only observed around specular directions with respect to the pigment surface. In coatings based on interference pigments, the color shift is mainly important when varying the incidence angle, while keeping constant a low aspecular angle (angle between viewing and specular directions), because at high aspecular angles, most of the special effect pigments do not reflect in the viewing direction, prevailing the color of the common diffusively reflecting absorption pigments [7–14]. An example of this color shift with the geometry is shown in the CIE a*,b* diagram [30] in Figure 1, for one of the highly goniochromatic interference-based special effect coatings studied in Reference [9] (Colorstream® T20-04 WNT Lapis Sunlight). For this figure and the following, the color coordinates at every geometry were calculated from the spectral BRDF measurements using the CIE-D65 illuminant and the CIE-1964 standard observer. Notice that, in this type of diagram, the closer to (0,0) the lower chroma, and that the tangent a*/b* relates with the colour hue. Solid lines represent those geometries with constant aspecular angles and different incidence angles, known as interference lines, since they show the impact of the interference pigments in the color. The longer the interference line, the lower the aspecular angle. Two similar interference lines are observed for each aspecular angle in the figure. One corresponds with the 'cis' and the other with the 'trans' aspecular direction [9]. Dashed lines represent geometries with constant incidence angle on the sample and different aspecular angle, having a roughly constant hue but different chroma, since towards larger aspecular angles the less chromatic color of the absorption pigments prevails. Given the characteristics of the color shift in these coatings, the more critical conditions for spectral BRDF measurements are those at constant and low aspecular angle, which will impose the measuring limitation in terms of the above-mentioned finite-size effect.

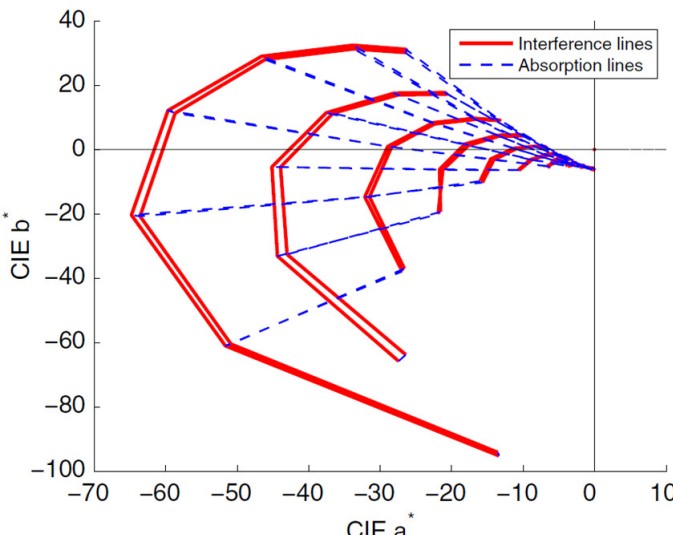

**Figure 1.** Color shift of a highly goniochromatic interference-based special effect coating (reprinted with permission from [9] © The Optical Society).

Diffraction pigments have a grating structure which deflects the incoming light. Like interference pigments, their optical properties are described by wave optics, but in this case their effect is not only observed around specular directions, but also at other directions. Since the deflection depends on the wavelength, the color at a given observation direction depends on the deflection angle with respect to the normal direction to the pigment. In coatings based on diffraction pigments, the color shift reveals very clearly when varying the aspecular angle at a constant incidence angle [2,15–17,31]. An example of this color shift with the geometry is shown as a CIE a*b* diagram in Figure 2, for one of the highly goniochromatic diffraction-based special effect coating studied in Reference [15] (SpectraFlair® Silver 1500-14). Again, solid lines represent those geometries with constant aspecular angles and different incidence angles, and it can be noticed that the hue variation is much smaller at constant aspecular angle than at constant incidence angle and varying aspecular angles, when it is possible to obtain all hues with roughly the same chroma for a given incidence angle. This makes the aspecular angle the most relevant angular variable when assessing the interrelation between spectral and angular variations and its impact in the spectral BRDF measurements of these coatings.

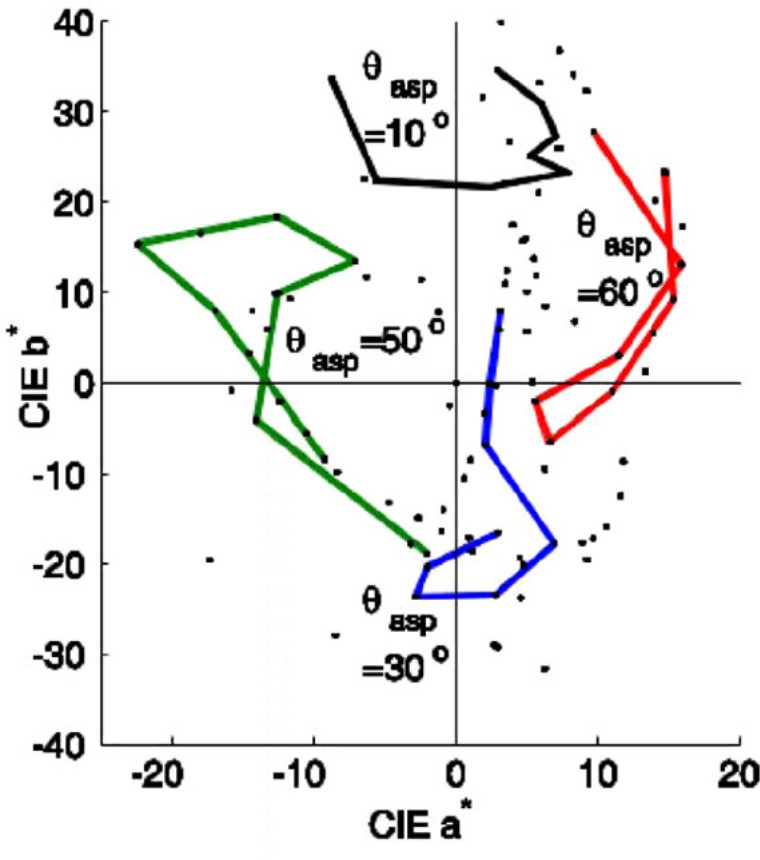

**Figure 2.** Color shift of a highly goniochromatic diffraction-based special effect coating (reprinted with permission from [15] © The Optical Society). Solid lines represent Bidirectional Reflectance Distribution Function (BRDF) measurements on the incidence plane, whereas dots represent out-of-plane measurements.

## 2. Materials

In previous articles, in collaboration with other authors, we studied the color travel of interference-based coatings [9,13,14] and diffraction-based coatings [15] of different manufactures from the measurement of the spectral BRDF at different geometries. For this work, we selected coatings showing very large hue change, which will impose the most restrictive conditions to be measured. They are those with Colorstream® T20-04 WNT Lapis Sunlight and Colorstream® T20-02 WNT Arctic

Fire, as interference-based coatings, and SpectraFlair® Silver 1500-35 and SpectraFlair® Plus 25, as diffraction-based coatings.

Coatings with SpectraFlair® pigments were produced by Viavi Solutions (SpectraFlair® is a registered trademark of Viavi Solutions, Inc., San Jose, CA, USA). The application is a solvent based paint reverse coated on PET film. All SpectraFlair® grades are made of magnesium fluoride and aluminum.

Coatings with Colorstream® pigments were produced by Merck KGaA ( Colorstream® is a registered trademark of Merck KGaA). They are spray coated with a conventional solvent based paint system on a black cardboard and include a clear coat on top. These Colorstream® pigments consist mainly of a $SiO_2$-substrate, coated with a $TiO_2$-layer.

## 3. Methodology

To estimate the error produced by non-infinitesimal spectral bandwidths ($\Delta\lambda$) and finite measurement solid angle values, denoted hereafter 'angular bandwidths', $\Delta\theta$, the spectral BRDF of highly goniochromatic special effect coatings was measured with the gonio-spectrophotometer thoroughly described in Reference [22], with a spectral bandwidth of 3 nm to 4 nm and an angular bandwidth lower than 1° both for irradiation and collection. Measurements were carried out within the incidence plane, covering the whole range of incidence and collection angles, both with an angular step of 10°. This angular step is good enough to interpolate at intermediate angles, at least for estimating typical angular and spectral variations for highly goniochromatric coatings. From those measurements, by using a spline interpolation, spectral BRDF data with an angular step of $(1/3)°$ were obtained, which will be regarded in this work as BRDF data with infinitesimal angular bandwidth. As an example of the spectral distributions used in this work, Figure 3a,b (interference- and diffraction-based coatings, respectively) show the spectral BRDF data corresponding to Colorstream® T20-04 WNT Lapis Sunlight (a) and SpectraFlair® Silver 1500-14 (b). Every solid line corresponds to a different geometrical configuration: fixed aspecular angle ( $\theta_{asp} = 10°$) and values of the angle of incidence $\theta_i$ from 10° to 70° (angular step of $(1/3)°$) for the interference-based special effect coating; while fixed $\theta_i = 10°$ and values of $\theta_{asp}$ from $-60°$ to 10° (angular step of $(1/3)°$) for the diffraction-based special effect coating.

These spectral curves correspond to a bandwidth of approximately 3 nm–4 nm, as indicated. The spectral bandwidth effect is well-known [32] and could be corrected by using some of the literature methods to obtain values equivalent to 1 nm bandwidth or strict values of the spectral distribution (infinitesimal bandwidth), but it is not necessary for the purpose of this work, where the measurement bandwidth effect over typical distributions is to be assessed. Therefore, in this work, the spectral curves shown in Figure 3a,b will be considered as testing distributions free of finite-size effect (infinitesimal-size-intervals BRDF).

The spectral or angular bandwidths of many commercial instruments are not known by their users. The shape of the spectral bandwidth can vary from a roughly isosceles triangle to a rectangle, passing through a trapezoid, and its spectral extension can also vary along the spectral interval of measurement. Since this work tries to give an estimate of the error, a rectangular bandwidth has been chosen, considering that it produces the worst results in many cases. An analogous rationale can be followed for the collection solid angle of instruments. Generally, these solid angles have an angular distribution that follows a cosine potential law, so assuming a rectangular-shaped angular distribution can also give the largest error that can be committed for a given sample.

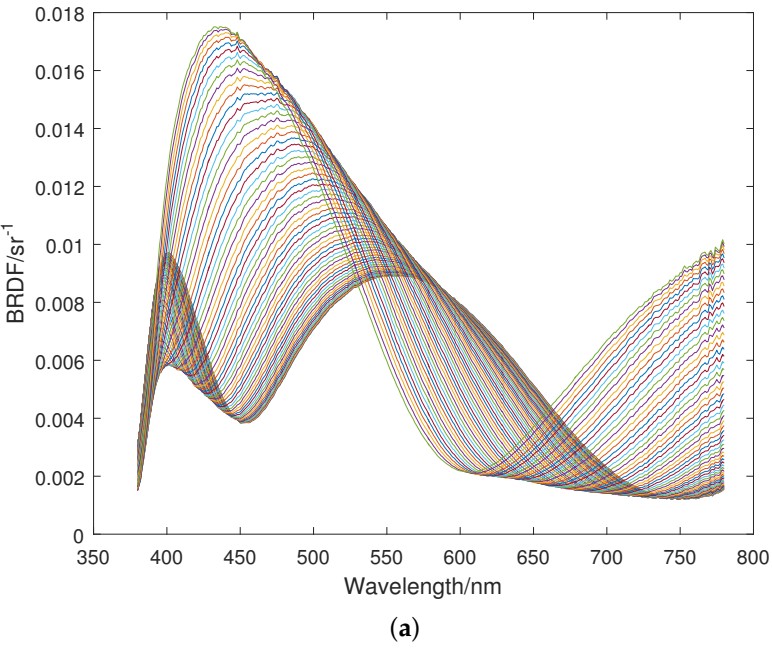

(**a**)

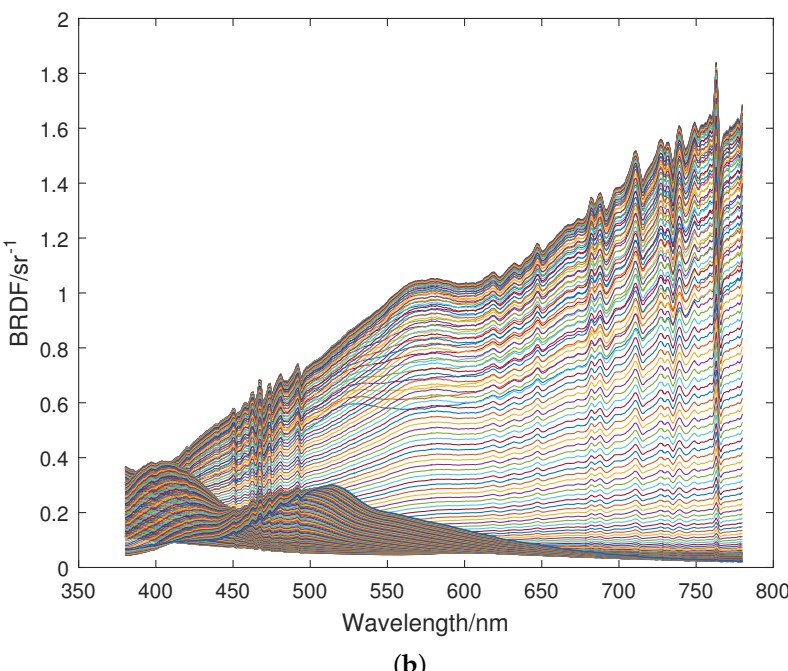

(**b**)

**Figure 3.** Interpolated spectral BRDF data for a highly goniochromatic (**a**) interference- and (**b**) diffraction-based special effect coating. The spectral BRDFs correspond to geometries with a fixed $\theta_{asp} = 10°$ and values of $\theta_i$ from $10°$ to $70°$ (angular step of $(1/3)°$) in (**a**), and to geometries with a fixed $\theta_i = 10°$ and values of $\theta_{asp}$ from $-60°$ to $10°$ (angular step of $(1/3)°$) in (**b**); (**a**) Interference-based special effect coating (Colorstream® T20-04 WNT Lapis Sunlight); (**b**) Diffraction-based special effect coating (SpectraFlair® Silver 1500-14).

From these data, the effect of applying larger spectral ($\Delta\lambda$) and angular ($\Delta\theta$) bandwidths is estimated by numerical calculation, up to 19 nm and 6.3°, respectively, by simply averaging values within the extended spectral and angular range, as:

$$\text{BRDF}(\lambda, \theta; \Delta\lambda, \Delta\theta) = \frac{\sum_{\lambda_i = \lambda - \frac{\Delta\lambda}{2}}^{\lambda + \frac{\Delta\lambda}{2}} \sum_{\theta_i = \theta - \frac{\Delta\theta}{2}}^{\theta + \frac{\Delta\theta}{2}} \text{BRDF}(\lambda_i, \theta_i)}{\sum_{\lambda_i = \lambda - \frac{\Delta\lambda}{2}}^{\lambda + \frac{\Delta\lambda}{2}} \sum_{\theta_i = \theta - \frac{\Delta\theta}{2}}^{\theta + \frac{\Delta\theta}{2}} 1} \tag{1}$$

where $\text{BRDF}(\lambda_i, \theta_i)$ is the infinitesimal-size-intervals BRDF, and $\theta$ is the relevant angle.

As explained in the previous section, the relevant angular variable for assessing the interrelation between spectral and angular variations depends on the optical phenomenon producing the colour change. For interference-based special effect pigments, the greatest color shift is observed by varying the incidence angle while keeping a low aspecular angle, which is the longest line shown in Figure 1. Therefore, the incidence angle at a low aspecular angle is the relevant variable, and the variation of the spectral BRDF with the incidence angle at a aspecular angle of 10° was examined.

In the case of diffraction-based special effect pigments, the largest colour shift covering even different hue areas is observed by varying the aspecular angle for a given fixed incidence angle (see Figure 2). Therefeore, the aspecular angle for a given fixed incidence angle is the most relevant variable, and the variation of the spectral BRDF with the aspecular angle at an incidence angle of 10° was examined.

Relative standard deviations of these spectral BRDFs for larger spectral and angular bandwidths with respect to the one assumed as "infinitesimal-size-intervals" are calculated as:

$$\epsilon_r(\lambda, \theta; \Delta\lambda, \Delta\theta) = \left| 1 - \frac{\text{BRDF}(\lambda, \theta; \Delta\lambda, \Delta\theta)}{\text{BRDF}(\lambda, \theta)} \right| \tag{2}$$

It provides, for each bandwidth pair ($\Delta\lambda$, $\Delta\theta$), distributions of relative errors across all possible measurement pairs ($\lambda$, $\theta$). Four examples of these distributions (small-small, large-small, small-large and large-large spectral and angular bandwidths) are given for the interference- and diffraction-based special effect coatings, in Figure 4a,b, respectively.

The finite bandwidth effect is quantified by calculating two descriptors, one based on the relative errors of the BRDF measurement at given wavelength ($\lambda$) and angle ($\theta$), and other based on the colour differences at each geometry.

The BRDF-error descriptor for a given pair ($\Delta\lambda$, $\Delta\theta$) is defined so that it represents the relative error at a confidence level of 95%, and therefore it is calculated as the percentile 95 of the relative errors across all measurement pairs ($\lambda$, $\theta$), $P_{95}(\epsilon_r)$.

The colour-difference descriptor for a given pair ($\Delta\lambda$, $\Delta\theta$) is defined so that it represents the CIEDE2000 colour-difference, currently recommended by CIE and ISO for pairs with colour differences below 5 CIELAB units [33], at a confidence level of 95%. It is calculated as the percentile 95 of this colour difference $\Delta E_{00}$ across all geometries, $P_{95}(\Delta E_{00})$. $\Delta E_{00}$ have been calculated for the reference conditions: Illuminant D65, CIE 1964 standard colorimetric observer and parametric factors $k_L$, $k_C$ and $k_C$ equal to unity. Calculations of colour coordinates have been done with a 1 nm summation step to avoid any bias [34].

The goniochromatism of the coatings here used as references is among the highest in coatings of the same type, so that the results here presented will seldom underestimate the error committed by the examined finite-size effect.

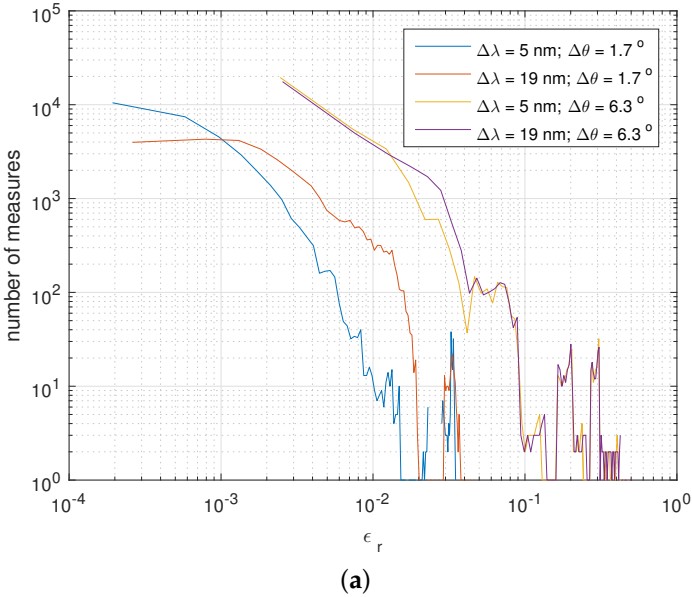

(**a**)

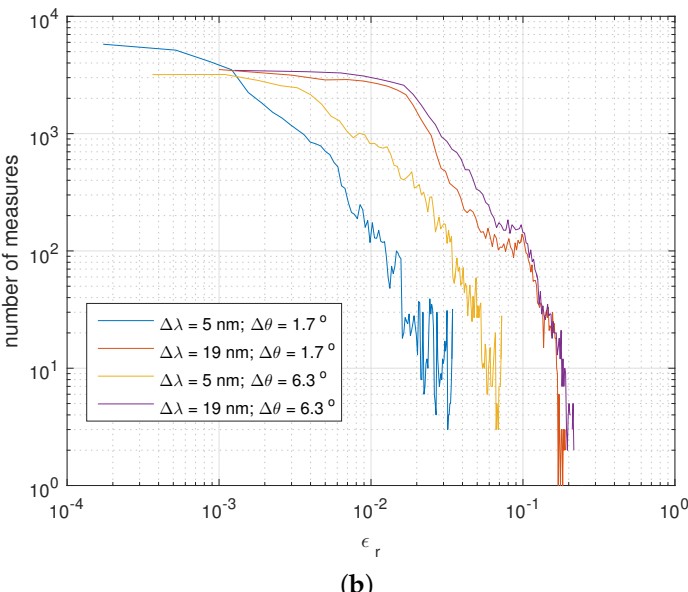

(**b**)

**Figure 4.** Distribution of Spectral-BRDF-error for the smallest and largest values of spectral and angular bandwidths used in this work.(**a**) Interference-based special effect coating; (**b**) Diffraction-based special effect coating.

## 4. Results and Discussion

The spectral-BRDF-error descriptor values, $P_{95}(\epsilon_r)$, obtained in this study for interference- and diffraction-based special effect coatings are shown in Figure 5a,b, respectively, for the pairs of bandwidths ($\Delta\lambda$, $\Delta\theta$) considered in this study. Results for two highly goniochromatic coatings of each type are represented, identified by solid and dashed lines. It is verified that the trend of the results does not depend on the specific sample.

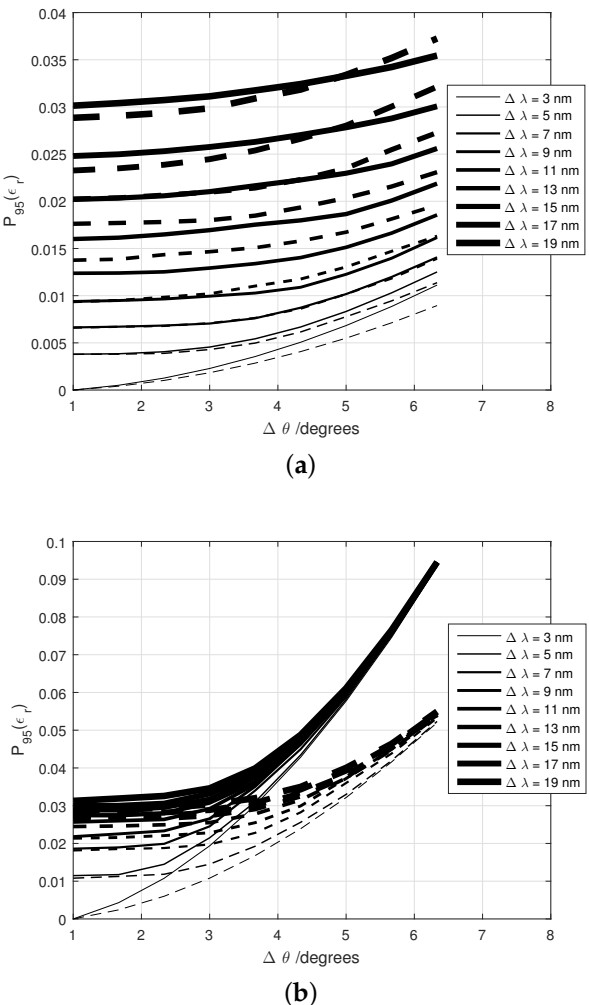

**Figure 5.** Spectral-BRDF-error descriptors, $P_{95}(\epsilon_r)$, obtained in this study for (**a**) Interference-based special effect coatings (Colorstream® T20-04 WNT Lapis Sunlight (solid line) and Colorstream® T20-02 WNT Arctic Fire (dash line)) (**b**) Diffraction-based special effect coatings (SpectraFlair® Silver 1500-35 (solid line) and SpectraFlair® Plus 25 (dash line)). Two highly goniochromatic coatings of each type are represented, identified by solid and dashed lines.

For diffraction-based special effect coatings, $P_{95}(\epsilon_r)$ increases much more with the angular bandwidth than with the spectral bandwidth. With the spectral bandwidth, it increases fast in the beginning, reaching a saturation level at around 10 nm. With respect to the angular bandwidth, no saturation is observed, growing slowly at the beginning and more at the end of the studied range.

When comparing the two types of coatings in Figure 5, it is observed that the impact of the angular bandwidth on the error of the BRDF values is greater for diffraction-based coatings than for interference-based coatings. Furthermore, the impact of both finite bandwidths ($\Delta\lambda$ and $\Delta\theta$) is similar for the two samples in the case of interference-based coatings, which is not the case for diffraction-based coatings when using large angular bandwidths. This result is also important to be considered to design instruments to measure special effect coatings.

It is observed in Figure 5 that the impact of the value of $\Delta\theta$ is much larger for diffraction-based special effect coatings than for interference-based, whereas the impact of the spectral bandwidth $\Delta\lambda$ is comparable.

Similarly, the colour-difference-BRDF-error descriptors, $P_{95}(\Delta E_{00})$, obtained in this study for interference- and diffraction-based special effect coatings are shown in Figure 6a,b, respectively, for the selected pairs of bandwidths ($\Delta\lambda$, $\Delta\theta$). Again, the same highly goniochromatic coatings of each type

are represented, identified by solid and dashed line. Thus, it can be verified that the trend of the results does not depend on the specific sample.

As shown in Figure 6a, $P_{95}(\Delta E_{00}) = 0.35$ CIEDE2000 units for the studied interference-based coatings, well under the just-noticeable difference of 1, at least up to the examined spectral and angular bandwidths. This colour difference is only larger than 1 when $\Delta\theta$ is larger than 18°.

In the case of diffraction-based coatings, Figure 6b, it is observed that $P_{95}(\Delta E_{00})$ is much larger than for interference-based coatings and that depends much more on the angular than on the spectral bandwidth. The results in Figure 6b show that colour differences above 1 CIEDE2000 units can be found for $\Delta\theta$ values larger than 4°, regardless the value of $\Delta\lambda$. Therefore, narrow angular bandwidths are paramount for colorimeters intended to assess the color appearance of diffraction-based coatings.

From these results, the recommendations shown in Table 1 can be given for the maximum values of spectral bandwidth and angular bandwidths that measurement instruments must have, in order to avoid exceeding the indicated levels of uncertainty in the BRDF measurement.

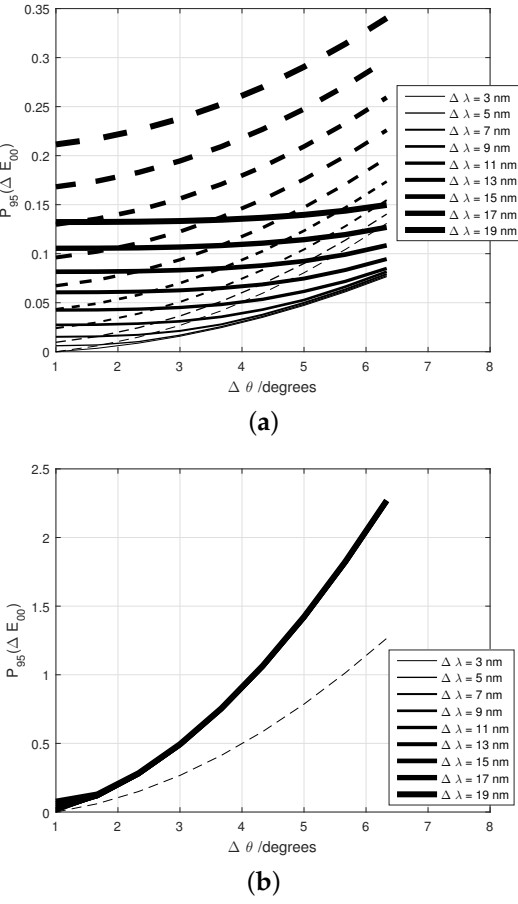

**Figure 6.** Colour-difference descriptors, $P_{95}(\Delta E_{00})$, obtained in this study for (**a**) interference- and (**b**) diffraction-based special effect coatings. Two highly goniochromatic coatings of each type are represented, identified by solid and dashed line (**a**) Interference-based special effect coatings (Colorstream® T20-04 WNT Lapis Sunlight (solid line) and Colorstream® T20-02 WNT Arctic Fire (dash line)); (**b**) Diffraction-based special effect coatings (SpectraFlair® Silver 1500-35 (solid line) and SpectraFlair® Plus 25 (dash line)).

**Table 1.** Recommendations on spectral and angular bandwidths for measuring the spectral BRDF of goniochromatic coatings based on interference and diffraction pigments.

| Recommended Angular/Spectral Bandwidths | Target Relative Uncertainty (k = 2) |
|:---:|:---:|
| **Goniochromatism Based on Interference Pigments** | |
| $\leq 4°$/3 nm | <0.5% |
| $\leq 5°$/7 nm | <1% |
| $\leq 6°$/11 nm | <2% |
| $\leq 6°$/17 nm | <3% |
| **Goniochromatism based on diffraction pigments** | |
| $\leq 2°$/3 nm | <1% |
| $\leq 3°$/5 nm | <2% |
| $\leq 3°$/11 nm | <3% |

## 5. Conclusions

The impact of using non-infinitesimal spectral and angular bandwidths has been assessed for spectral BRDF measurements of goniochromatic surfaces, considering a set of interference- and difraction-based coatings. Instrument designers should pay attention to the balance between spectral and angular bandwidths when optimizing the signal to noise ratio, because their impact on the measurement is not equivalent. From the analysis, recommendations have been given for the measurements of goniochromatic samples. The diffraction-based samples are more dependent on the angular bandwidth than those based on interference. The results indicate that it is convenient to standardize these bandwidths. Recommended values have been given for different levels of uncertainty target. For a similar uncertainty value, spectral and angular bandwidth requirements are stricter for diffraction-based coatings than for interference-based ones. Defining indicators based on the 95th percentile of the BRDF error or of the color difference is a useful procedure to establish criteria and recommendations on the thresholds for variables of influence on the measurement of BRDF or color. The quality control of goniochromatic paints in automotive, cosmetic, packaging or other sectors should benefit from these recommendations, since they allow noticeable color differences to be avoided through good practice or better optimized instruments.

**Author Contributions:** Conceptualization, A.F.; methodology, A.F.; formal analysis, A.F.; resources, J.C.; data curation, A.F.; writing–original draft preparation, A.F.; writing—review and editing, A.F. and J.C.; project administration, J.C.; funding acquisition, J.C. All authors have read and agreed to the published version of the manuscript.

**Funding:** This research was funded by EMPIR 16NRM08 Project "Bidirectional reflectance definitions" (BiRD), and by Comunidad de Madrid, grant number S2018/NMT-4326-SINFOTON2-C. The EMPIR is jointly funded by the EMPIR participating countries within EURAMET and the European Union.

**Acknowledgments:** We want to thank Alfred Schirmacher (Physikalisch-Technische Bundesanstalt, Germany) for his comments and corrections of the manuscript.

**Conflicts of Interest:** The authors declare no conflict of interest.

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
