# Peer review of "Angular and Spectral Bandwidth Considerations in BRDF Measurements of Interference- and Diffraction-Based Coatings"

_coatings, doi:10.3390/coatings10111128_

Round 1

Reviewer 1 Report

In equation 1 the sum of denominator should be corrected replacing 1 with i.

Author Response

In equation 1, the sum of denominator should be corrected replacing 1 with i.

***Authors: We do not agree. The summation in the denominator represents the number of points in order to calculate the average value as explained in the paragraph above the equation. Therefore, writing number one and adding it as many times as indicated by the summation steps and its limits, the number of points to be averaged will be obtained.

Reviewer 2 Report

By reading this manuscript, I see that there is little content that may relate to the topic of the journal, i.e., coatings.

The authors make a thorough presentation and discussion related to the BRDF but this would be more appropriate for a spectroscopy or physics journal.

The authors do not mention in detail what are the materials that were used for testing. The spectroscopy part is well described, but the coating/surface science part lacks detail (in fact it is absent). On which type of materials/coatings were the measurements performed? What are the details of the experimental setup (regarding the material component, not the optical acquisition)?

The authors should present their measurements in correlation to different coatings obtained in different configurations/colors and to discuss the influence of coating type/color(pigment)/thickness/roughness on the spectral parameters. So, each instance of testing should be correlated to a well described system (material) with a certain (known) composition.

Also, the introduction lacks the description of the actual importance of the study from a materials/surface properties point of view. why is your approach relevant for this journal?

I do not recommend the publication of this manuscript in Coatings, lest the authors provide a solid anchoring of their study to an actual well-defined material or system.

Author Response

By reading this manuscript, I see that there is little content that may relate to the topic of the journal, i.e., coatings.

The authors make a thorough presentation and discussion related to the BRDF but this would be more appropriate for a spectroscopy or physics journal.

***Authors: The reviewer thinks the work described in this manuscript is not close to the scope of the journal. However, the journal has recently prepared a special number and we were invited to contribute to it by one of the guest editors of that number. We did not contribute then because we could not find enough time to meet the deadline.

The authors do not mention in detail what are the materials that were used for testing. The spectroscopy part is well described, but the coating/surface science part lacks detail (in fact it is absent). On which type of materials/coatings were the measurements performed? What are the details of the experimental setup (regarding the material component, not the optical acquisition)?

***Authors: Interference- and diffraction-based special effect coatings are used in this work as stated in the text, and in our opinion this the only description relevant for this study. The paper is about how to measure the reflectance properties of representative coatings of these kinds, the BRDF generally speaking, and what measurement parameters are more relevant and so have to be more carefully considered in the assessment. Specific materials were not described in detail but the kind of materials can be tracked through the references, if it is needed.  Since reflectance properties essentially depend on the material type, two typical materials (interference- and diffraction-based coatings) with very large color shift were selected. Nevertheless, the specific materials used are identified in the reviewed version of the manuscript (see Fig. 3, 4 and 5, and lines 49, 70 and 90), although the results of the work do not vary for other materials of the same kind.

The authors should present their measurements in correlation to different coatings obtained in different configurations/colors and to discuss the influence of coating type/color(pigment)/thickness/roughness on the spectral parameters. So, each instance of testing should be correlated to a well described system (material) with a certain (known) composition.

***Authors: Measurements are presented in correlation to different coating types, but not in correlation to the parameters defining each type (thickness and so on), because the aim of this work is not discussing the influence of the parameters, but the influence of measurement parameters in the goodness of the result. The study proposed by the reviewer is another completely different study. The aim of this work was to provide general recommendation for color and BRDF measurements of goniochromatic coatings, a previous stage to study the influence of structural parameters on spectral parameters.   

Also, the introduction lacks the description of the actual importance of the study from a materials/surface properties point of view. why is your approach relevant for this journal?

***Authors: The importance of the study from a materials/surface properties point of view is that the goodness of the characterization depends on a proper selection of the measurement parameters.

I do not recommend the publication of this manuscript in Coatings, lest the authors provide a solid anchoring of their study to an actual well-defined material or system.

Reviewer 3 Report

line 24 - change impacts and affects to impact and affect to match plural noun

line 83 - I believe you want to insert "spectral BRDF data" between the corresponding

line 84 and Fig. 3 - what the specific geometric configurations are that correspond to each line should be defined

line 173 - extra comma

line 174 - change Instruments to Instrument

Nice paper!

Author Response

Line 24 - change impacts and affects to impact and affect to match plural noun

***Authors: The correction was done in the new version.

Line 83 - I believe you want to insert "spectral BRDF data" between the corresponding

***Authors: Right. The correction was done in the new version.

Line 84 and Fig. 3 - what the specific geometric configurations are that correspond to each line should be defined

***Authors: We have specified them in the new version by adding the following sentences:

In line 91-93:

“Every solid line corresponds to a different geometrical configuration: fixed θasp=10º and values of θi from 10º to 70º for interference based coating, while fixed θi =10º and values of θasp from −60º to 10º for diffraction-based coatings

And in the caption of Fig. 3.

  “The spectral BRDFs correspond to geometries with a fixed θasp=10º and values of θi from 10º to 70º (angular step of (1/3)º) in (a), and to geometries with a fixed θi =10º and values of θasp from −60º to 10º (angular step of (1/3)º) in (b).”

Line 173 - extra comma

***Authors: The correction was done in the new version.

Line 174 - change Instruments to Instrument

***Authors: The correction was done in the new version.

Reviewer 4 Report

Given below are my comments to improvise the quality of the manuscript.

  1. Please explain what is the novelty in this work, how this work is different from the ones available in the literature.
  2. Line 84: Which geometrical configurations, please explain.
  3. Line 104: How can you say that the angle of incidence is the most relevant variable for the discussed case? What is the inference from the large color shift?
  4. Line 106: Explain how the aspecular angle is the most reliable variable for the diffraction-based special effect pigments.
  5. Line 165: Colour differences above 1CIEDE2000 units can be expected for Δθ values greater than 4 degrees. How? (Please explain)
  6. Please add the utility/application of your work in industry in section 4.

Author Response

Given below are comments to improvise the quality of the manuscript.

  1. Please explain what is the novelty in this work, how this work is different from the ones available in the literature.

***Authors: We have added the following text after the second sentence, right before “Since the appearance of iridescent (or goniochromatic) surfaces… (lines 16-25)”:

“The spectral BRDF of goniochromatic surfaces changes in great extent for different combinations of illumination and viewing directions (geometries), making their colorimetric description complex. Most of the research work on the appearance of goniochromatic surfaces has dealt with the proper methodology to describe their color by selecting adequate geometries, and to use measurements for rendering such surfaces. However, the metrological aspect of measuring the spectral BRDF and the color with real instruments, which have finite-size apertures, has not been systematically studied yet as done in this work.”

  1. Line 84: Which geometrical configurations, please explain.

***Authors: We have specified them in the new version by adding the following sentences:

In line 91-93:

“Every solid line corresponds to a different geometrical configuration: fixed θasp=10º and values of θi from 10º to 70º for interference based coating, while fixed θi =10º and values of θasp from −60º to 10º for diffraction-based coatings

And in the caption of Fig. 3.

  “The spectral BRDFs correspond to geometries with a fixed θasp=10º and values of θi from 10º to 70º (angular step of (1/3)º) in (a), and to geometries with a fixed θi =10º and values of θasp from −60º to 10º (angular step of (1/3)º) in (b).”

  1. Line 104: How can you say that the angle of incidence is the most relevant variable for the discussed case? What is the inference from the large color shift?

***Authors: We wrote in the new version (112-115):

“For interference-based special effect pigments, the greatest color shift is observed by varying the incidence angle while keeping a low aspecular angle, which is the longest line shown in figure 1. Therefore, the incidence angle at a low aspecular angle is the relevant variable, the variation of the spectral BRDF with the incidence angle at a aspecular angle of 10º was examined.”

  1. Line 106: Explain how the aspecular angle is the most reliable variable for the diffraction-based special effect pigments.

***Authors: We wrote in the new version (116-119):

“… the largest colour shift covering even different hue areas is observed by varying the aspecular angle for a given fixed incidence angle (see Fig. 2). Therfeore, the aspecular angle for a given fixed incidence angle is the most relevant variable, and the variation of the spectral BRDF with the aspecular angle at an incidence angle of 10º was examined.”

  1. Line 165: Colour differences above 1CIEDE2000 units can be expected for Δθ values greater than 4 degrees. How? (Please explain)

***Authors: Probably the expression is not clear. We meant that, according to the results shown in Fig. 6b, colour differences above 1 CIEDE2000 units appear for Δθ values greater than 4 degrees. We wrote in the new version (176-177):

“The results in Fig. 6b show that colour differences above 1 CIEDE2000 units can be found for ∆θ values larger than 4º.

  1. Please add the utility/application of your work in industry in section 4.

***Authors: We added the following sentence at the end of section 4 (195-197):

“The quality control of goniochromatic paints in automotive, cosmetic, packaging or other sectors should benefit from these recommendations, since they allow noticeable color-differences to be avoided through good practice or better optimized  instruments.”

Round 2

Reviewer 4 Report

Thank you for incorporating my suggestions. The overall quality of the manuscript is much improved now.

Author Response

Thank you for the revision. It contributed for the improving of the manuscript.